# Theoretical and empirical quantification of the accuracy of polygenic scores in ancestry divergent populations

Ying Wang[1], Jing Guo[1], Guiyan Ni[1], Jian Yang[1,2], Peter M. Visscher[1] & Loic Yengo[1]✉

Polygenic scores (PGS) have been widely used to predict disease risk using variants identified from genome-wide association studies (GWAS). To date, most GWAS have been conducted in populations of European ancestry, which limits the use of GWAS-derived PGS in non-European ancestry populations. Here, we derive a theoretical model of the relative accuracy (RA) of PGS across ancestries. We show through extensive simulations that the RA of PGS based on genome-wide significant SNPs can be predicted accurately from modelling linkage disequilibrium (LD), minor allele frequencies (MAF), cross-population correlations of causal SNP effects and heritability. We find that LD and MAF differences between ancestries can explain between 70 and 80% of the loss of RA of European-based PGS in African ancestry for traits like body mass index and type 2 diabetes. Our results suggest that causal variants underlying common genetic variation identified in European ancestry GWAS are mostly shared across continents.

[1] Institute for Molecular Bioscience, The University of Queensland, Brisbane QLD 4072, Australia. [2] Institute for Advanced Research, Wenzhou Medical University, Wenzhou, Zhejiang 325027, China. ✉email: l.yengodimbou@uq.edu.au

Polygenic scores (PGS, also known as PRS when applied to diseases) are now routinely utilised to predict complex traits and risk of diseases from findings of genome-wide association studies (GWASs). Over recent years, the predictive performances of PGS have steadily increased with GWASs sample sizes, as predicted by theory[1]. However, the over-representation of European ancestry in the majority of GWASs has been shown to yield an unbalanced improvement of PGS prediction accuracy in non-European ancestry populations[2,3]. For example, Duncan et al.[2] report the average accuracy of PGS across multiple traits to be ~64% lower in individuals of African ancestry as compared with that in individuals of European ancestry. Similarly, Martin et al.[3] report, across multiple traits, reductions of PGS accuracy of ~37%, ~50% and ~78% in individuals of South-Asian, East-Asian and African ancestries, respectively, relative to individuals of European ancestry.

Although increasingly emphasised in the recent GWAS literature, it is worth noting that the loss of accuracy problem is not utterly new. Indeed, a number of studies in the animal breeding literature have previously reported lower accuracy of genomic selection across genetically distant breeds[4,5], consistent with the observation of limited transferability of GWAS findings across diverse human populations[6,7]. These studies also highlight major factors influencing that loss such as differences between populations in causal variants effect sizes, in alleles frequencies and in linkage disequilibrium (LD) between causal variants and SNPs assayed in GWAS[6,8,9]. To illustrate the latter point, let us consider a SNP which has an LD $r^2$ with a causal variant of 0.8 in the discovery population and 0.6 in the target population. Such a SNP would therefore explain $25\% = (1 - 0.6/0.8)$ less trait variation and thus be less predictive in the target population as compared with the discovery population, even when causal variants and their effect sizes are shared between ancestries. More generally, previous empirical and simulation studies have shown that accuracy of genetic predictors decays monotonically with increased genetic differentiation ($F_{ST}$) and LD differences between ancestries[4,7,10]. Other factors such as population specific causal variants[11,12], gene × environment interaction[13,14] have been implicated as potential explanations of the loss of PGS prediction accuracy.

In addition, deterministic formulas have been derived to predict the accuracy of genomic prediction across breeds as a function of population parameters (e.g. heritability, genetic correlation) and also using selection index theory[15,16]. However, these deterministic formulas mostly apply to best linear unbiased predictors[17], which are not classically used in human studies. Consequently, a theoretical understanding of the *trans*-ancestry predictive capacities of standard PGS is still missing. Although the key factors causing the loss of accuracy have been enumerated in previous studies, quantification of their relative contributions has not been done systematically. Note that quantifying the relative contributions of all these factors is critical for understanding aetiological differences between ancestries, which may have important clinical implications. Here, we develop an approximation of the theoretical relative accuracy of PGS in an ancestry divergent sample as a function of population genetics parameters. Our method only requires GWAS summary statistics and ancestry-specific reference panels. We evaluate the performances of our theory through extensive simulations and apply it to GWAS of 5 quantitative traits and 3 common diseases with different genetic architectures in ~350,000 unrelated UK Biobank participants.

## Results

### Expected relative accuracy of PGS in ancestry divergent populations.
We consider a quantitative trait $y$, for which the genetic component is underlain by random additive effects of $M_C$ causal variants. Without loss of generality, we assume causal variants to be shared between ancestries but allow their effect sizes to vary from one ancestry to another. Therefore, ancestry-specific causal variants are a special case with non-zero effect sizes in only one ancestry. We then assume that a GWAS of $y$ has been performed in a discovery sample of a given ancestry, hereafter denoted by Population 1 and that a PGS, defined as the sum of minor allele counts weighted by their estimated effects from the discovery GWAS, is used to predict $y$ in a target sample of another ancestry, hereafter denoted by Population 2. The study design is shown in Supplementary Fig. 1.

We derived the expected accuracy of such PGS in Population 2 (denoted $R_2^2$) as function of the expected accuracy in a sample of same ancestry as Population 1 (denoted $R_1^2$), the minor allele frequencies (MAF) $p_{k,1}$ and $p_{k,2}$ at the $k$th PGS-SNP (i.e. SNPs included in the PGS) in Populations 1 and 2, respectively, the LD between the $j$th causal SNP and the $k$th PGS-SNP in Population 1 and 2, respectively (denoted $r_{jk,1}$ and $r_{jk,2}$), the heritabilities $h_1^2$ and $h_2^2$ of $y$ in Populations 1 and 2, respectively, and the correlation $\rho_b$ of causal SNP effects between Population 1 and Population 2. It is worth underlining here that direct attempts to predict $R_1^2$ or $R_2^2$ are challenging as they require prior knowledge of the number of causal variants ($M_C$). Unfortunately, no method to date can provide estimates of $M_C$ with high enough precision. However, under the assumption that causal variants are shared between ancestries, focusing on the ratio $R_2^2/R_1^2$ overcomes this limitation and therefore allows us to derive the approximate closed-form formula shown below in Eq. (1) (details of our derivations are given in Supplementary Note 1):

$$R_2^2/R_1^2 \approx \frac{\rho_b^2 h_2^2}{h_1^2} \times \left( \frac{\sum_{k=1}^{M_T} \sqrt{\frac{p_{k,2}(1-p_{k,2})}{p_{k,1}(1-p_{k,1})}} \left[ \sum_{j=1}^{M_C} r_{jk,1} r_{jk,2} \right]}{\sum_{k=1}^{M_T} \left( \sum_{j=1}^{M_C} r_{jk,1}^2 \right)} \right)^2 \times \frac{\mathrm{var}(\mathrm{PGS}_1)}{\mathrm{var}(\mathrm{PGS}_2)},$$

(1)

where $M_T$ denotes the number of GWS SNPs used to calculate PGSs. Note that a special case of Eq. (1) was derived in de Vlaming et al.[18] to characterize the accuracy of PGS in the presence of causal effects heterogeneity (modelled in their work by a parameter denoted $\rho_G$ akin to our parameter $\rho_b$) between cohorts of the same ancestry.

Equation (1) shows that the relative accuracy (RA, relative to the accuracy in populations of same ancestry as Population 1) of PGS, defined as $R_2^2/R_1^2$, can be discomposed as the product of multiple terms: (i) the squared genetic correlation (the correlation of effect sizes of causal variants) between populations, (ii) the ratio of heritability between populations, (iii) the ratio of squared covariances between PGS and $y$ in both populations (approximated by the product of the first two terms on the right hand side of Eq. (1); Supplementary Note 1); and (iv) the ratio of variance of PGS in both populations. This decomposition allows us to distinguish and thus separately quantify the fraction of the RA that is attributable to differences in effect size distribution between populations (term (i) and term (ii) in the decomposition, including $\rho_b^2$, $h_2^2$ and $h_1^2$) and the fraction attributable to alleles frequencies and LD differences between populations (term (iii) and term (iv) in the decomposition). It is important to underline that the contribution of each of these factors can differ between traits.

Many terms in Eq. (1) can be quantified a priori using information from previous studies or from reference panels. However, the big unknown in Eq. (1) remains the LD between unobserved causal variants and PGS-SNPs. Understanding how much PGS-SNPs tag causal variants is critical to quantify and therefore predict the RA of PGS. To reduce this uncertainty, we

focus in this study on PGS based on independent genome-wide significant (GWS) SNPs, which are more informative of the location of causal variants than sub-significant SNPs. Importantly, previous studies[2,3,19,20] have shown reduced predictive performance of PGS across ancestry when the PGS includes sub-significant SNPs, which provides an additional rationale for concentrating on GWS SNPs. Note also that in the near future, as GWAS sample sizes increase, the accuracy of GWS-based PGS will become similar to that of genome-wide PGS approaches.

Given that causal variants are largely unknown, we propose a heuristic method that considers as a candidate causal variant, any SNP in LD ($r^2 > 0.45$) with a GWS SNP and located within 100 kb of the latter (Supplementary Fig. 2). This heuristic is justified by a previous study by Wu et al.[21] which has quantified the fine-mapping precision of GWAS and has found over multiple computer simulations that causal variants lied within 100 kb of the GWS SNPs ~90% of the time and that LD $r^2$ between causal and GWS SNPs was >0.45. Once candidate causal variants are identified for each independent GWS included in the PGS, we approximate Eq. (1) by replacing $r_{jk,l}^2$ and $r_{jk,1}r_{jk,2}$ with the average of these quantities over all candidate causal variants, as shown below in Eq. (2):

$$R_2^2/R_1^2 \approx \frac{\rho_b^2 h_2^2}{h_1^2} \times \left( \frac{\sum_{k=1}^{M_T} \overline{r_{k,1}r_{k,2}} \sqrt{\frac{p_{k,2}(1-p_{k,2})}{p_{k,1}(1-p_{k,1})}}}{\sum_{k=1}^{M_T} \overline{r_{k,1}^2}} \right)^2 \times \frac{\sum_{k=1}^{M_T} p_{k,1}(1-p_{k,1})\hat{\beta}_k^2}{\sum_{k=1}^{M_T} p_{k,2}(1-p_{k,2})\hat{\beta}_k^2},$$

(2)

where $\hat{\beta}_k$ represents the effect size of the $k$th PGS-SNP estimated in the discovery GWAS. We use the notation $\overline{r_{k,1}^2}$ to denote the mean squared correlation of allele counts between the $k$th PGS-SNP and all candidate causal SNPs within 100 kb. Similarly, we define $\overline{r_{k,1}r_{k,2}}$ as the mean of $r_{jk,1}r_{jk,2}$'s between the $k$th PGS-SNP and all candidate causal SNPs within 100 kb.

Moreover, we assume that the accuracy of the PGS in samples of same ancestry as the discovery GWAS is known (e.g. estimated in an independent sample) and propose to quantify other input parameters such as allele frequencies and LD correlations using data from an ancestry diverse reference panel like that of Phase 3 of the 1000 Genomes Project (1KGP)[22]. Finally, the quantification of $\rho_b$ and $h^2$ requires access to phenotypic and genotypic data from the target population, which may not be available simultaneously. Therefore, when such data are unavailable our method can only quantify the fraction of the RA that is explained by allele frequencies and LD differences between populations.

**Performance of the method on simulated data.** We ran computer simulations to evaluate the performances of Eqs. (1) and (2) under various genetic architectures. We also assessed the performances of a naive approach that assumes GWS SNPs to be the causal variants. In this case the expected RA explained by allele frequencies and LD differences between populations would approximately equal to $\frac{1}{M_T} \left( \sum_{k=1}^{M_T} \sqrt{\frac{p_{k,2}(1-p_{k,2})}{p_{k,1}(1-p_{k,1})}} \right)^2 \times \frac{\text{var}(\text{PGS}_1)}{\text{var}(\text{PGS}_2)}$. When PGS-SNPs are independent and given that SNP effect sizes from GWAS are typically small and of similar magnitude, this naive approach can be further considered as a function of the ratio of heterozygosity at GWS SNPs between ancestries.

Our simulations utilise existing genotypes at ~1.1 million common HapMap3 SNPs imputed in 351,983 unrelated UK Biobank (UKB) participants. These participants were categorised into four ancestry homogeneous groups corresponding to

European ancestry (EUR; $N_{EUR} = 333,263$), East-Asian ancestry (EAS; $N_{EAS} = 2257$), South-Asian ancestry (SAS, $N_{SAS} = 9448$) and African ancestry (AFR; $N_{AFR} = 7015$). The European ancestry group was further divided into a discovery set of $N = 313,284$ participants in which GWS SNPs were identified (Methods), a validation set in which the accuracy of PGS within-European-ancestry was quantified and a reference group in which we predicted the accuracy of PGS. A thorough description of how these groups were defined is given in the Methods section. As our main focus is to predict the fraction of the RA that can be attributed to alleles frequencies and LD differences between populations, we therefore assumed that effect sizes of causal variants are perfectly correlated across populations, i.e. $\rho_b = 1$ and that heritability is constant across populations, i.e. $h_2^2 = h_1^2 = h^2$. Note that in practice these assumptions are likely to be violated. However, our theoretical framework (Eqs. (1, 2)) allows us to study the contribution of MAF and LD and that of heritability and genetic correlation separately. We relax this assumption below. In total, we simulated six scenarios corresponding to three values of the number of causal variants $M_C = 1000$, 5000 and 10,000; and two values of trait heritability $h^2 = 0.25$ and 0.5 (Methods).

As expected, we observed across all scenarios that accuracies of PGS decreased monotonically with increased genetic distance to EUR (Supplementary Fig. 3). The genetic distance was measured as $F_{ST}$ described in Supplementary Note 2. More specifically, we found the largest RA in individuals of Asian ancestry (mean RA ~91% in SAS and mean RA ~77% in EAS), which has an average $F_{ST}$ of ~0.06 with EUR. The smallest RA was observed in individuals of AFR ancestry (~46%), which has an average $F_{ST}$ of ~0.14 with EUR (Supplementary Figs. 3 and 4). These results imply that *trans*-ancestry predictive power of PGS remains limited even when causal variants and their effect sizes are shared between ancestries[20,23]. Across 18 simulation scenarios and over 100 simulation replicates for each scenario (Fig. 1), we found differences between the mean observed RA and the predicted RA from Eq. (1) to range between −4.5 and +2.5%. Although these differences were statistically significant in 11/18 simulation scenarios (two-tailed $t$-test, $p$-value <0.05/18), we found their sign to be inconsistent between ancestries. In fact, predicted RA from Eq. (1) slightly underestimates the observed RA in SAS and AFR ancestries but yields a small overestimation in EAS ancestry. Consequently, we found on average over all non-EUR ancestries, that Eq. (1) produces unbiased predictions, i.e. not statistically different from the observed RA (two-tailed $t$-test, $p$-value = 0.46). Similarly, we found our heuristic approach based on candidate causal variants to yield unbiased predictions of the RA in 7/18 simulated scenarios. More specifically, differences across scenarios between the mean observed RA and that predicted from Eq. (2) ranged between −5.3 and +5.2%. Note that this range is larger than when using information on causal variants. On average across simulation scenarios, we found that the approach assuming GWS SNPs to be the causal variants (referred to RA$_{pred3}$ in Fig. 1) strongly overestimated the RA in non-European ancestries. The average overestimation relative to the observed RA ranged from +4.3% (i.e. (96.3%−92.3%)/92.3%× 100%)) in SAS up to +103.0% in AFR ancestry. This result suggests that population differences in LD between causal variants and GWS SNPs contribute a larger fraction to the decreased RA than allele frequencies differences at GWS SNPs only. It is also worth noting that our predictions were nearly insensitive to using either whole-genome sequence (WGS) data from the 1KGP or imputed genotypes of UKB participants as reference panels (Supplementary Fig. 5), which is reflected by the highly correlated allele frequencies (Supplementary Fig. 6) and

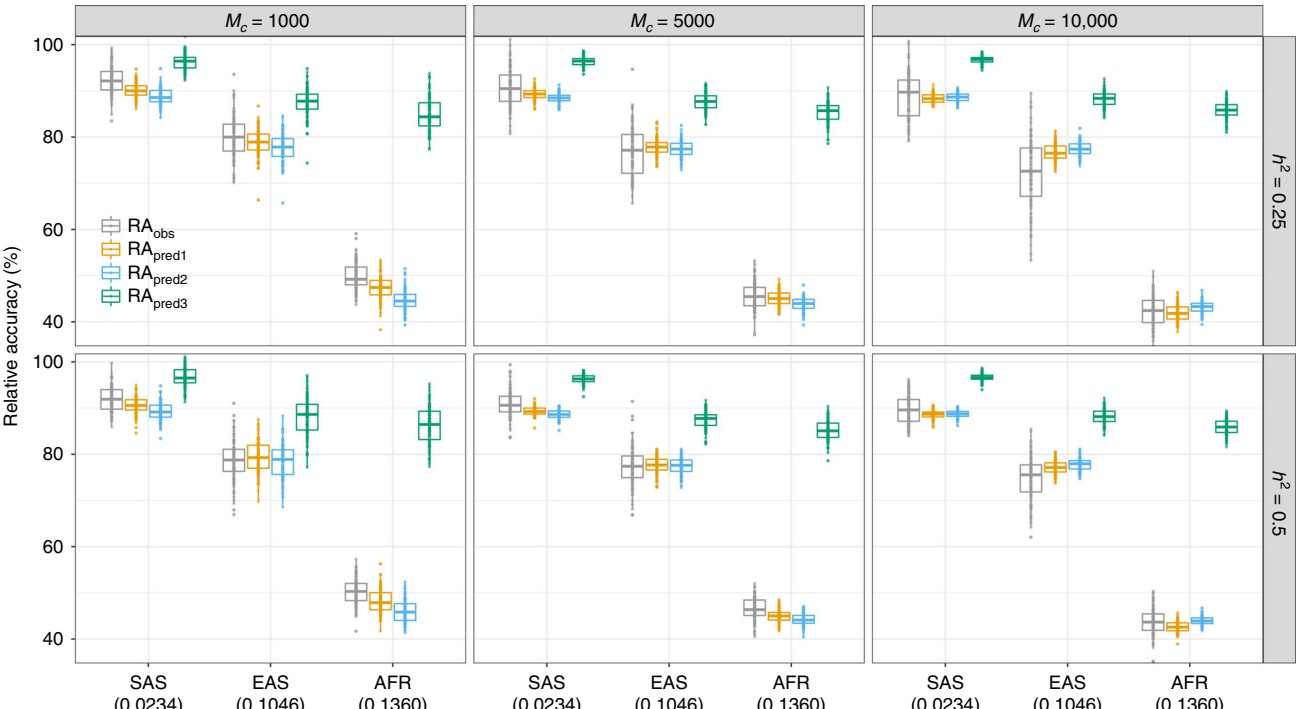

**Fig. 1 *Trans*-ancestry relative prediction accuracy of PGS in different simulation scenarios.** Relative accuracies (RA) were calculated as the ratio of the squared correlation between PGS and simulated trait in UKB participants of non-European ancestry over the same squared correlation estimated in 10,000 independent UKB participants of European ancestry (Methods). We varied trait heritability ($h^2 = 0.25$ and 0.5) and numbers of $M_C$ causal variants ($M_C = 1000$, 5000 and 10,000) in the simulations. RA$_{obs}$ refers to the observed RA calculated. The predicted RA labelled as RA$_{pred1}$ is estimated using Eq. (1) based on parameters calculated from SNP pairs of PGS-SNPs and known causal variants within 100 kb; RA$_{pred2}$ refers to RA calculated using SNP pairs of PGS-SNPs and candidate causal variants using Eq. (2). RA$_{pred3}$ refers to the naive predicted RA using Eq. (1) when assuming that PGS-SNPs are the causal variants. The numbers under the ancestry labels in x-axis denoted the pairwise $F_{ST}$ calculated using HapMap3 SNPs between discovery population and target population (see Supplementary Note 2). Boxes represent the first and third quantiles and whiskers are 1.5-folds the interquartile range. The points represent the RA for 100 replicates. The median estimates are shown as the horizontal line in the boxes.

LD scores (Supplementary Fig. 7) between WGS and imputed data (Supplementary Note 3). However, LD reference panels with larger sample sizes are still recommended to achieve more accurate estimates of LD correlations. Finally, we assessed the robustness of our results by varying the size of the discovery GWAS between 100,000 and 300,000 participants (Supplementary Note 4). We found the accuracy of PGS to increase proportionally in all ancestries, such that the RA remained constant and thus independent of sample size (Supplementary Fig. 8).

Altogether, our simulation results show the validity of our theory and highlight its ability to predict with little bias (<5%) the RA attributable to allele frequencies and LD differences between ancestries under various scenarios.

**Impact of negative selection.** In addition to heritability and polygenicity, another important aspect of the genetic architecture of complex traits and diseases is the relationship between effect sizes at causal variants (hereafter denoted by $\beta$) and their minor alleles frequencies (hereafter denoted by $p$). This relationship has been modelled in many studies[24–28] using a parameter $S$ such that $\beta^2$ is assumed to be proportional to $[2p(1-p)]^S$. Values of $S$ determine the relative contributions of common versus rare variants to the genetic variance in the population and thus have been used as an indirect measure of the strength of natural selection[29]. Our model assumes that the variance explained by each causal SNP is constant regardless of allele frequencies. This assumption is consistent with a strong negative selection on

causal variants shared between populations and corresponds to a value of $S = -1$. Although previous studies[24,26,30] have reported pervasive negative selection on complex traits and diseases, these studies often report estimates of $S$ with less extreme magnitudes than that assumed in our model. Moreover, given that little is known on the strength of negative selection in non-European populations, we next investigated through additional simulations the impact of violations of this assumption.

We adopted a similar framework (Methods) as in our first simulation. However, for the sake of simplicity, we fixed the number of causal variants to $M_C = 5000$ and the trait heritability to $h^2 = 0.5$. We denoted $S_1$ and $S_2$ as the value of $S$ in Population 1 and Population 2, respectively. We considered three scenarios corresponding to (i) $S_1 = S_2 = -0.5$, i.e. equal strength of negative selection in both populations, (ii) $S_1 = -0.75$ and $S_2 = -0.5$, i.e. stronger selection in Population 1 and (iii) $S_1 = -0.5$ and $S_2 = -0.75$, i.e. stronger selection in Population 2.

In non-European ancestries, we found over 100 replicates that our theory based on Eq. (1) mostly predicts a smaller RA than actually observed on average in the three scenarios (Fig. 2). This therefore makes our approach conservative. We observed a slightly larger downward bias in the predicted RA when using our heuristic approach based on candidate causal SNPs. On average, the latter approach showed an absolute underestimation of the RA of ~2.0% in SAS ancestry (i.e. the difference between the observed and the predicted RA is 92%−90% = 2.0%), 6.5% in EAS ancestry and 8.5% in AFR ancestry, which thus provides a lower bound for the RA. Interestingly, we did not find significant differences in observed and predicted accuracies across scenarios

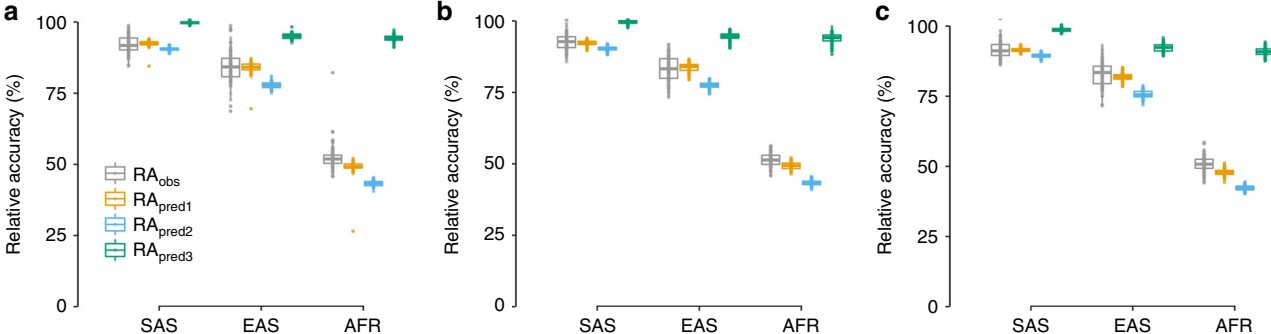

**Fig. 2 Impact of negative selection on PGS *trans*-ancestry relative accuracies.** Relative accuracies (RA) of PGS in different ancestries under various strengths of negative selection. Traits were simulated with a heritability $h^2 = 0.5$ and assuming $M_C = 5000$ causal variants. Negative selection was modelled using a parameter $S$ such that smaller values of $S$ indicate stronger strength of selection. Values of $S$ are denoted $S_1$ and $S_2$ in the discovery population and target populations, respectively. We considered thee scenarios: **a** $S_1 = S_2 = -0.5$; **b** $S_1 = -0.5$, $S_2 = -0.75$; and **c** $S_1 = -0.75$, $S_2 = -0.5$. RA_obs, RA_pred1, RA_pred2 and RA_pred3 labels are defined as in the legend of Fig. 1. Boxes represent the first and third quantiles and whiskers are 1.5-folds the interquartile range. The points represent the RA for 100 replicates. The median estimates are shown as the horizontal line in the boxes.

(ANOVA test, *p*-value >0.05). This somewhat surprising observation is suggestive that when heritability is constant and effect sizes of causal variants are perfectly correlated, differences in strengths of selection between ancestries might have a negligible impact on the RA of PGS. As a consequence, we can expect differences in strengths of selection between ancestries to mainly impact the term $\rho_b^2 h_2^2 / h_1^2$ in Eq. (1). Note, however, that our simulations were based on observed contemporary LD differences between ancestries, which have likely already been shaped by negative selection. This limitation may have masked an additional contribution of differential selection between ancestries.

**Application to real data.** We performed GWASs of 5 quantitative traits and 3 common diseases with different genetic architectures in 313,284 unrelated UKB participants of EUR ancestry (Supplementary Note 5). The 5 quantitative traits are standing height (Height), body mass index (BMI), HDL and LDL cholesterol (HDL and LDL) and triglycerides (TG); and the 3 common disease (cohort prevalence >5% in each ancestry) are asthma, type 2 diabetes (T2D) and hypertension (HTN). We report in Supplementary Table 1, the numbers of quasi-independent GWS SNPs for each trait and disease (Methods). We used these GWS SNPs to create polygenic predictors of each trait and disease then evaluated their predictive performances in the validation subsamples of the UKB as described in the Methods section. We evaluated the accuracy of PGS of diseases on the liability scale using ancestry-specific disease prevalence estimated in the UKB and the transformation proposed previously by Lee and collegues[31] (Methods). Note that using ancestry-specific prevalence from previous population studies[32–35] did not change our results (Supplementary Fig. 9). We also assessed the predictive accuracy of PGS based upon sub-significant SNPs (Supplementary Note 6) and found, in individuals of non-European ancestries, that PGSs including SNPs selected at less stringent *p*-value thresholds did not systematically improve over using GWS SNPs only (Supplementary Fig. 10). This important observation is consistent with previous studies[2,3,19,20] and emphasises that observed RA from *p*-value thresholding scoring methods can be seriously underestimated if based on sub-significant SNPs.

As previously reported[2,3], we found the average observed RAs across traits and diseases to decrease monotonically with increasing $F_{ST}$ from EUR ancestry. More specifically, the mean observed RA across traits and diseases is 72%, 64% and 24% in participants of SAS, EAS and AFR ancestry, respectively.

Consistently, the number of traits and diseases for which the reduction of RA was statistically significant (Wald test, *p*-value <0.05) also differed between ancestries (Supplementary Table 2). In SAS ancestry for example, the reduction of RA was significant only for height, BMI, LDL and HDL. However, in EAS ancestry the reduced RA of TG and Asthma PGS was also significant, while PGSs of all traits had significantly reduced RA in AFR ancestry.

Despite this monotonically decreasing trend on average across traits, we found the observed RA of the LDL PGS to be larger in participants of EAS (RA = 58%; standard error S.E. 10%) and AFR (RA = 40%; standard error S.E. 5.0%) ancestries as compared with participants of SAS ancestry (RA = 35%; S.E. 4.1%). We further investigated this observation and found this pattern to be explained by a single large effect variant (namely rs7254892 with an estimated SNP effect $\hat{\beta} = 0.49$ standard deviation per allele, MAF estimated in 1KGP = 0.03, 0.04, 0.07 and 0.16 in EUR, SAS, EAS and AFR ancestries, respectively) which explains 1.5% of LDL variance in participants of EUR ancestry. Excluding this variant from the LDL PGS led to a reduced accuracy in participants of EAS and AFR ancestries (RA = 30%; S.E. 7.8% for EAS and RA = 16%; S.E. 3.3% for AFR; Supplementary Fig. 11) below that from participants of SAS ancestry (RA = 37%; S.E. 4.5%). This result is important as it shows how the distribution of SNP effects can lead to unexpected patterns of RA, and therefore challenge straightforward comparisons between traits with different genetic architectures.

We next applied our method to predict the fraction of RA attributable to MAF and LD differences between ancestries (Fig. 3). We used WGS data from the 1KGP as a reference panel for LD and allele frequency calculations. For traits with significantly reduced RA, we define the loss of accuracy (LOA) of their corresponding PGS as LOA = $(1 − RA) \times 100\%$. We also define the proportion of LOA explained by LD and MAF as the ratio between the predicted LOA over the observed LOA (Supplementary Fig. 12). The derivation of standard errors of this proportion is given in Supplementary Note 7. Our method predicts a different contribution of LD and MAF across traits and ancestries (Supplementary Table 2, Fig. 3). For example, we predict that MAF and LD differences between EUR and SAS ancestries explain ~24% (S.E. 2%) of the LOA of the height PGS, while we predict those factors to explain ~38% and ~72% of the LOA in EAS and AFR ancestry, respectively. For T2D and HTN, respectively, we found that ~82% (S.E. 14%) and ~86% (S.E. 20%) of the LOA can be expected because of MAF and LD differences

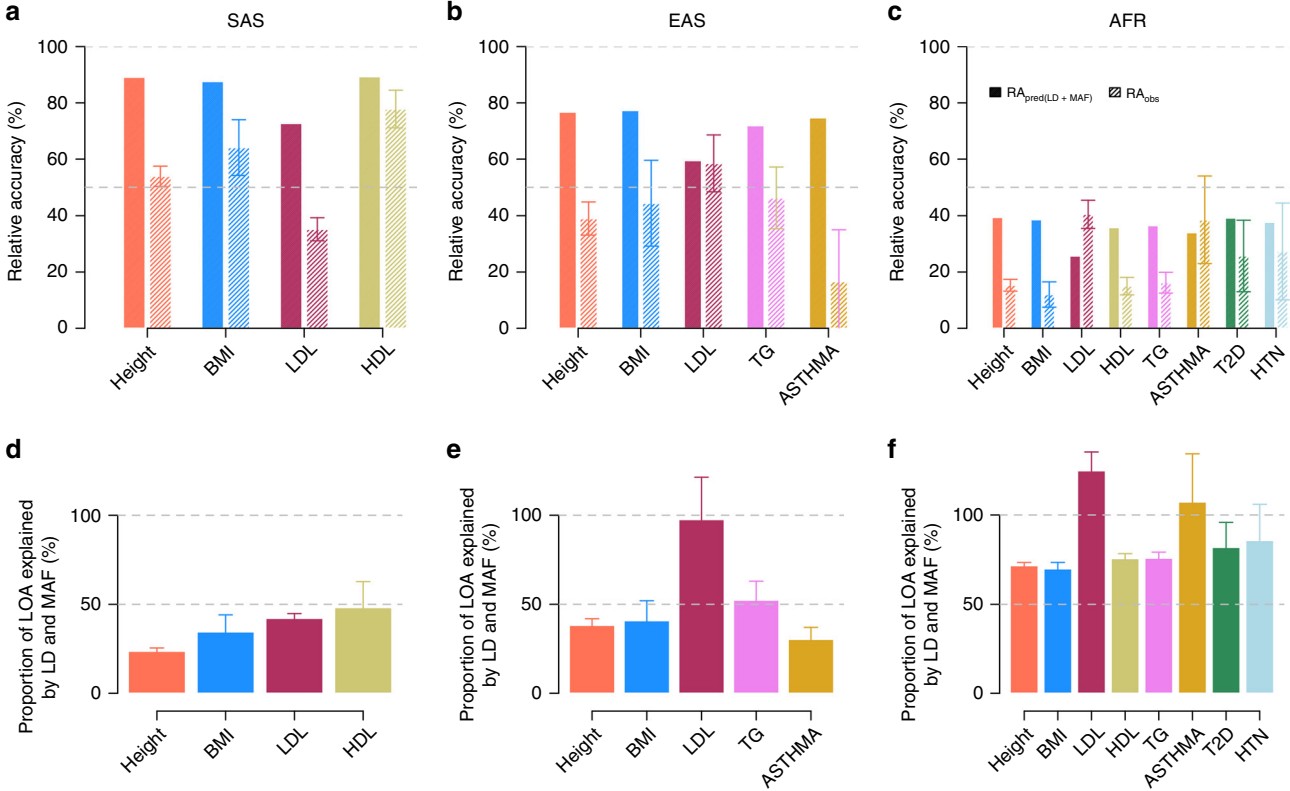

**Fig. 3 Trans-ancestry relative prediction accuracy of PGS of 5 quantitative traits and three common diseases. a–c** Relative accuracies (RA) are calculated as the ratio of the squared correlation between PGS and traits/diseases in UKB participants of non-European ancestry over the same squared correlation estimated in ~20,000 independent UKB participants of European ancestry. We report here only ancestry-trait/disease pairs, with a significant reduction in RA (Wald test, p-value <0.05). Data for all ancestry-trait/disease pairs are provided in Supplementary Table 2. $RA_{pred(LD+MAF)}$ refers to the RA predicted from Eq. (2) only using information from LD and MAF differences between ancestries. $RA_{obs}$ refers to observed RA calculated using independent genome-wide significant trait-associated SNPs. Panels **d–f** show the proportion of the loss of accuracy (LOA) explained by LD and MAF (Supplementary Fig. 12) calculated as $100\% \times (1 - RA_{pred(LD+MAF)})/(1 - RA_{obs})$. The grey dashed lines are $y = 100\%$ and $y = 50\%$. Error bars in the figure represent the standard errors of observed RA or proportion of LOA explained by LD and MAF in each ancestry-trait/disease pair, which calculation is detailed in Supplementary Note 7.

between EUR and AFR ancestries. However, we note that the standard errors of the latter predictions are large. On average over traits (with a significantly reduced RA), we found that MAF and LD differences alone can explain up to ~37% (S.E. 5%), ~57% (S.E. 11%) and ~86% (S.E. 7%) of the LOA in SAS, EAS and AFR ancestry, respectively. Standard of average over traits were calculated using leave-one-trait-out jackknife (Supplementary Note 7).

Altogether, our results show that MAF and LD can explain >50% of the loss of accuracy of PGS in individuals of AFR or EAS ancestry, while the RA of PGS in SAS is likely dominated by other factors such as heritability differences or low correlation of causal effects.

## Discussion

In this study, we developed a new theory to predict the relative prediction accuracy of PGS across populations of different ancestries. Our theory overcomes the challenge of predicting accuracy directly by modelling the relative prediction accuracy instead. Under the assumption that causal variants are shared but allowing their effects to differ between populations, we have shown that the contribution of MAF and LD differences at causal SNPs and that of other parameters of the genetic architecture (heritability and genetic correlation, i.e. $\rho_b$) can be quantified separately. We have shown through simulations that the

contribution of differences in LD and MAF to the RA of PGS can be predicted with little bias (<5%) using a simple heuristic approach that models local correlation of LD and MAF across ancestries in the close vicinity of GWS SNPs. Our approach only requires GWAS summary statistics as well as data from a globally-diverse reference panel such as the 1KGP.

We explored the impact of negative selection through additional simulations by considering different relationships between effect sizes and allele frequencies of causal variants. We found that predictions from our method can underestimate the RA by up to −8.5% in populations undergoing mild negative selection (Fig. 2). Therefore, our predictions can be interpreted as upper bounds for how much RA can be reduced simply because of MAF and LD differences between ancestries. More generally, since PGSs in the current study are mainly focused on common variants, the impact of low frequency and rare variants which may better inform selection remains to be investigated.

We further assessed the ability of our theory to predict the accuracy of PGS of multiple traits and diseases in non-European UKB participants. Altogether, we found that between ~70% (S.E. 7.0%) and 100% (S.E. 20.6%) of the reduction of RA of PGS in AFR ancestry could be explained by differences in LD and MAF. Importantly, we found that the fraction of the RA attributable to MAF and LD varied between traits, which mostly reflects differences in the genetic architecture (heritability, polygenicity and cross-ancestry effect size correlation: $\rho_b$) of these traits. It is

noteworthy that AFR participants of the UK Biobank reside in the UK and therefore are likely to share similar environments as EUR participants. As consequence, the relative contribution of $\rho_b$ and $h^2$, which partially reflects the effects of gene by environment interactions (if any), might be underestimated. Similarly, the contribution of $\rho_b^2 h_2^2/h_1^2$ to the RA of PGS in individuals of SAS and EAS might be even larger if evaluated across continents than reported in this study. A recent study by Durvasula et al.[12] suggested that ~50% of the heritability is captured by European specific variants when a trait is under moderate negative selection, thus limiting the upper bound of prediction accuracy in AFR for example when using European-derived GWAS summary statistics. Our results based on common variants also show a RA <50% in AFR but we demonstrate that this reduction is mostly explained by MAF and LD differences at common causal variants and not necessarily by population specific causal variants. The latter conclusion is further supported by large genetic correlations observed between EUR and non-EUR ancestries[36], which overall suggests that causal variants underlying common genetic variation identified in European ancestry GWASs are mostly shared across continents.

We note a few limitations to our study. First, our model uses LD and MAF estimated from a reference panel. We have shown the effectiveness of this strategy in population of homogenous ancestry; however, its application remains challenging in admixed populations with complex LD patterns and demographic history. In addition, our theoretical approach cannot be directly applied to predict the RA of PGS using SNP effects estimated from a *trans*-ancestry meta-analysis, although a straightforward extension could be derived for fixed-effects *trans*-ancestry meta-analyses. In practice, random effect models, which already accounts for allele frequency differences between ancestries, are often preferred over fixed-effects models. Therefore, we acknowledge that further theoretical work is required to address that specific question. Secondly, we only analysed common SNPs (with MAF >0.01 in each ancestry) in all populations, which limits the generalization of our conclusions to rarer variants. Indeed, rare variants are more likely to be population specific and are usually poorly imputed using a small imputation reference panel. Thirdly, our study has focused on PGS calculated from GWS SNPs alone while other methods such as LDpred[37] or SBayesR[38], which utilise information from the entire genome, have been shown to be more accurate within ancestry. Although we acknowledge that characterizing the theoretical RA of PGS based on genome-wide methods like LDpred or SBayesR deserves further investigation, we also emphasise that the gap in prediction accuracy between GWS SNPs and genome-wide methods is destined to shrink as the sizes of GWAS continue to grow. Nevertheless, we re-analysed our simulated data using these two methods (Supplementary Note 8) and found that their predictive performance relative to our clumping strategy is near proportional in EUR and non-EUR ancestries. Therefore, we see a similar RA across all methods (Supplementary Fig. 13) although SBayesR shows the largest RA across scenarios. The latter observation mirrors our simulation results showing a constant RA as sample size increases (Supplementary Fig. 8). Fourthly, our predictions of the contribution of LD to the RA of PGS can in principle be inflated in the presence of epistatic interactions between causal variants if they are in strong LD or if causal effect sizes are a function of local LD differences between populations. Lastly, although our simple heuristic strategy to identify candidate causal variants worked well in simulations, we expect the use of standard fine-mapping tools to further improve the efficiency of our method. More specifically, fine-mapping posterior probabilities could be used as weights for candidate causal SNPs, which our current heuristics cannot do. An advantage of our heuristic method is that it utilises whole-genome sequencing data and therefore candidate causal variants that are not present in the GWAS may still be used for inference. In contrast, standard fine-mapping tools are limited by the resolution of GWAS summary statistics, although that resolution can be improved using summary statistics imputation[39].

In conclusion, despite the acknowledgement of the necessity to collect large scale genome-wide data across different ancestries to fulfil the potential use of PGS in the precision medicine era[3], this goal remains difficult to achieve in the near future. Instead, *trans*-ancestry studies have been increasingly popular. They incorporate genotype data from different ancestries to boost statistic power with increasing sample sizes, which have the benefit to discover disease/trait-associated loci and fine-mapping causal variants associated with complex traits or diseases[13,40–43]. However, the structure of the reference population still remains to be thoroughly explored, such as whether some specific populations with certain sample sizes are mostly useful in *trans*-ancestry studies. Our model presents an opportunity for such study design using both the LD and allele frequency information in a population level. By performing *trans*-ancestry GWASs, we expect that the predictive ability would increase when the admixed LD structure and allele frequency of the discovery population is similar to the target population.

## Methods

**Samples and quality controls.** The UK Biobank (UKB) comprises of ~500,000 individuals recruited from the UK, aged from 40 to 69 years old. Participants were genotyped using two genotyping arrays, the Affymetrix UK BiLEVE Axiom™ Array and UK Biobank Axiom™ Array. Each participant provided written informed consent. The North West Multi-Centre Research Ethics Committee (MREC) approved the study and all participants in the UKB study analysed here provided written informed consent. Additional study and quality control details are shown in Bycroft et al.[44]. The approach to infer the ancestry of each individual is described in Yengo et al.[45]. We firstly projected each individual of UKB onto the genotypic principal components (PCs) calculated in 2000 participants of 1KGP[22]. We only extracted individuals from four ancestries in the 1KGP, namely, European ancestry (EUR, $N = 503$), South-Asian ancestry (SAS, $N = 489$), East-Asian ancestry (EAS, $N = 504$) and African ancestry (AFR, $N = 504$). We excluded African Caribbeans in Barbados (ACB) and Americans of African Ancestry in SW USA (ASW) populations from AFR and all individuals of American ancestry (AMR) considering their complex admixture patterns. We then assigned each of those genotyped participants of UKB to the closest ancestry based on the first three PCs, resulting in 463,795 EUR, 11,906 SAS, 2486 EAS and 9184 AFR. To remove cryptic relatedness in the UKB, we used the GCTA software to calculate the genomic relationship matrix (GRM)[46] based on genotyped SNPs in each of the aforementioned populations. With one of each pair of individuals with estimated relatedness larger than 0.05 being removed, a subset consisting of unrelated individuals was generated in each ancestry. For the European ancestry, we only extracted those self-reported British and Irish participants. After randomly sampling 10,000 individuals from British subset, we created the discovery dataset using the remaining 313,284 individuals. As for the target populations, we used an independent dataset of ~39,000 UKB individuals. Those individuals included the 10,000 randomly sampled participants who identified themselves as British, 9979 participants of EUR who identified themselves as Irish, the 9448 participants of SAS, the 2257 participants of EAS and the 7015 participants of AFR. Data from the 1KGP were used as the reference panel in this study. We generated subsets of unrelated individuals in 1KGP with the same strategy as described above, resulting in 495 EUR, 457 SAS, 498 EAS and 484 AFR.

For participants of non-European ancestries in the UKB, we further imputed the SNP array data to the 1KGP given that the imputation reference panels, Haplotype Reference Consortium (HRC)[47] and UK10K[48], used in the UKB are predominant by European descents thus a large number of missing SNPs are observed when using hard call (<0.1) thresholds on dosage data. In each ancestry we firstly extracted genotyped SNPs such that Hardy–Weinberg equilibrium (HWE) $p$-value >0.001 and missing rates <0.05 and also excluded individuals with genotype call rates <0.9. Filtered SNPs in each ancestry were then phased using SHAPEIT2[49] and imputed to 1KGP by IMPUTE2[50]. In each ancestry, stringent quality control procedures were performed separately. We removed SNPs with imputation quality scores <0.30, MAF < 0.01, HWE $p$-value <$10^{-6}$, or missing genotype call rates >0.05. HapMap3 SNP set, which has been well designed for human genome-wide common genetic variants[51], was then extracted from imputed data to run follow-up analyses. A total of 990,395 filtered HapMap3 SNPs in common between populations were selected.

SNPs in the 1KGP reference panel were restricted to a total of 6,877,707 SNPs common to all four ancestries after excluding those SNPs with minor allele count (MAC) <5 in each ancestry. The above described HapMap3 SNP set was further intersected with 1KGP, thus limiting the number to 978,783.

**Simulations**. We derived a deterministic equation to quantify the relative prediction accuracy of PGS across ancestries. To evaluate this equation, we performed various simulations (each scenario with 100 replicates) using the real genotypes in the UKB cohort. To simplify our notations, we denoted the training/discovery sample as Population 1 ($l = 1$) and the target/test sample as Population 2 ($l = 2$).

The phenotypes were simulated based on the additive model $y = g + e$ in all scenarios using different values of $M_C$ and trait heritability ($h^2$). In each simulation replicate, we simulated a trait with a heritability of 0.25 or 0.5 in all ancestries. Traits were simulated from $M_C$ ($M_C = 1000, 5000$ and $10,000$) causal variants sampled at random from the HapMap3 SNPs. We assumed the effect sizes of causal variants ($\beta$) were perfectly correlated across populations, i.e. $\rho_b = 1$. For each causal variant, $\beta$ was sampled from a normal distribution with mean 0 and variance $\frac{h^2}{2p_{jl}(1-p_{jl})M_C}$, where $p_{jl}$ is the MAF in $j$th causal variant in population $l$. For each individual, the genetic value $g$ was defined such as $g = \sum_{j=1}^{M} x_{jl}\beta_j$, where $x_{jl}$ denotes the minor allele count ($x_{jl}$ equals to 0, 1 or 2) at the $j$th causal variant in population $l$. The environmental effect ($e$) was simulated using a normal distribution with mean 0 and variance equal to $(1 - h^2)$: $e \sim N(0, 1 - h^2)$, such that the phenotypic variance across populations was equal to 1.

PLINK1.90 (note version 20 Mar was used in this study)[52] was used to run GWAS for the simulated phenotypes in Population 1 using the simple linear association testing on HapMap3 SNPs. To mimic the imperfect LD between the causal variants and the SNP markers used in GWAS, the causal variants were always left out of the analysis.

To further explore the impact of negative selection, we sampled $\beta$ from a multi-normal distribution with mean 0 and variance $2p_{jl}\left(1-p_{jl}\right)^S \sigma_\beta^2$, where $\sigma_\beta^2$ is the variance of causal effect sizes. We considered three scenarios corresponding to (i) equal strength of selection in both ancestries ($S_1 = S_2 = -0.5$), (ii) stronger selection in Population 1 ($S_1 = -0.75$ and $S_2 = -0.5$) and (iii) stronger selection in Population 2 ($S_1 = -0.5$ and $S_2 = -0.75$). The phenotypes were generated in the same way as described above. For simplicity, we focused on a trait with a heritability $h^2 = 0.5$ and controlled by $M_C = 5000$ causal variants.

**Selection of genome-wide significant (GWS) SNPs**. After running GWAS, we selected approximately independent SNPs associated with the trait (referred to as GWS SNPs or PGS-SNPs), using the LD clumping algorithm implemented in PLINK1.90[52]. We used the following command: --clump-p1 5e-8 --clump-p2 5e-8 --clump-kb 2000 --clump-r2 0.01. The genotypes of the training population were used as LD reference for clumping. We used here a more stringent LD threshold than classically used (e.g. 0.1 or 0.2) because SNPs with LD $r^2$ as large as 0.1 can still reflect the same signal when GWAS sample size is large (e.g. $N > 300,000$).

**Deterministic accuracy of PGS in *trans*-ancestry genetic prediction**. We calculated the deterministic accuracy in the target populations based on selected GWS SNPs. Since we focused on in this study the contribution of LD and allele frequency differences between populations to the RA, the term $\rho_b^2 h_2^2/h_1^2$ equalled to 1 in the simulations. We then used a reference panel to calculate the LD correlation and MAF between populations. To first validate our theory, we used Eq. (1), which assumes causal variants to be known, to calculate the LD correlation and MAF using SNP pairs between PGS-SNPs and known causal variants. We then explored the performance of our heuristic method using Eq. (2), given that causal variants are typically unknown or unobserved. For that, we took advantage of the LD and MAF information between PGS-SNPs and candidate causal variants instead. Further, we applied a naive approach assuming PGS-SNPs to be the causal variants, thus mainly the allele frequency differences between populations would be captured using Eq. (1).

When using Eq. (1) with known causal variants, we firstly matched GWS SNPs to them to calculate LD correlation and allele frequencies between populations (results shown as RA$_{pred1}$ in the simulations). It was done by constraining the window centred at each GWS SNP as 100 kb and then selecting those pairs including known causal variants. This window was based on the report that ~95% top lead SNPs (with MAF >0.01) identified from GWASs are within 100 kb distance from the causal variants in European ancestry[21]. Although the causal variants are often unknown or unobserved in a classical GWAS, they are usually tagged by numerous SNPs. Therefore, we took advantage of the information regarding fine-mapping precision of GWAS studies and selected candidate causal variants as those SNPs in LD $r^2$ >0.45 with GWS SNPs and located within 100 kb window[21]. Those GWS SNPs and candidate causal variants pairs were then used in Eq. (2), with results referring to as RA$_{pred2}$ in the simulations. When assuming the PGS-SNPs as causal variants, we estimated the accuracies using Eq. (1) where the LD correlation was replaced with 1 (results denoted as RA$_{pred3}$ in the simulations). The LD correlations were estimated using PLINK1.90[52] ($-r$).

The final predicted parameters to evaluate predictive ability were calculated as the mean of the estimates across 100 replicates, respectively. To explore the impact of imputation on our model, we used both 1KGP WGS data and UKB imputed

data as the reference in the simulated data. We used the same approaches, except for the one assuming causal variants were known, to analyse GWAS summary statistics of complex traits and diseases in UKB (Supplementary Note 4). While, 1KGP WGS data were used as the reference panel since most causal variants might not be included in the summary statistics.

The relative accuracy (RA) was calculated as the ratio of $R_2^2/R_1^2$, where $R_1^2$ was the predicted prediction accuracy in the population with same ancestry of discovery population; and $R_2^2$ was the predicted prediction accuracy in other target populations. Note we have two target populations of EUR when calculating $R_1^2$ in the simulations, we used the British subset ($N = 10,000$) as the validation, in which the accuracy of PGS within-European-ancestry was quantified and a reference group (self-reported Irish with $N = 9979$), in which we predicted the accuracy of PGS, separately.

**Empirical accuracy of PGS in *trans*-ancestry genetic prediction**. After selecting approximately independent GWS SNPs using LD clumping, we then generated PGS in each target population by adding up the product of minor allele counts times effect sizes of GWS SNPs estimated from GWAS summary statistics. The prediction accuracy was then estimated using the squared correlation ($R^2$) between the true phenotypes and the PGS. For diseases, we calculated the liability-scale $R^2$ as described in Lee et al.[31]. We estimated the disease prevalence as the proportion of cases in each ancestry given that UK Biobank is a population-based study. We combined the two target populations of EUR ancestry ($N = 19,979$) to calculate the corresponding $R^2$ for traits/diseases in UKB. The empirical relative accuracies (RA) were calculated as the ratio of the $R^2$ in UKB participants of non-European ancestry over the same $R^2$ estimated in ~20,000 independent UKB participants of European ancestry.

**Reporting summary**. Further information on research design is available in the Nature Research Reporting Summary linked to this article.

## Data availability

This study makes use of genotype and phenotype data from the UK Biobank data under project 12505 (http://www.ukbiobank.ac.uk/). UKB data can be accessed upon request once a research project has been submitted and approved by the UKB committee. 1KGP data can be accessed through ftp://ftp.1000genomes.ebi.ac.uk/vol1/ftp/data_collections/1000_genomes_project/data.

## Code availability

Data and R scripts for generating the main figures can be found at https://github.com/loic-yengo/Code_for_Wang_et_al2020/. Other methods used are as follows: GCTA: http://cnsgenomics.com/software/gcta; PLINK: https://www.cog-genomics.org/plink2; IMPUTE2: https://mathgen.stats.ox.ac.uk/impute/impute_v2.html; SHAPEIT2: https://mathgen.stats.ox.ac.uk/genetics_software/shapeit/shapeit.html; LDpred: https://github.com/bvilhjal/ldpred; GCTB-SBayesR: https://cnsgenomics.com/software/gctb/.

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

## Acknowledgements

This research was supported by the Australian National Health and Medical Research Council (1113400) and the Australian Research Council (DP160102400, FT180100186, FL180100072 and DE200100425). The funders had no role in study design, data collection and analysis, decision to publish, or preparation of the manuscript. This research has been conducted using the UK Biobank Resource under project 12505. We thank all participants of the UK Biobank.

## Author contributions

L.Y. and P.M.V. conceived the study. L.Y. derived the theory. L.Y. and Y.W. designed the experiment. Y.W. prepared data and performed analyses under the assistance and guidance from J.G., G.N., J.Y., P.M.V. and L.Y. Y.W. and L.Y. wrote the paper with the participation of all authors. All authors reviewed and approved the final paper.

## Competing interests

The authors declare no competing interests.
