## [Peer Review File · Nature Communications]

Reviewers' comments:

Reviewer #1 (Remarks to the Author):

Authors conducted a simulation study empirically quantifying transferring accuracy polygenic risk score (PRS) among ancestry divergent populations. They introduced a model which assesses effects of minor allele frequency (MAF) spectra and linkage disequilibrium (LD) differences. The simulation study demonstrated that majority of the decreased accuracy of PRS between the populations were mostly explained by MAF and LD differences. The PRS of Europeans (EUR) were mostly transferrable to South Asians (SAS) or East Asians (EAS), when compared to Africans (AFR), reflecting distances between the populations. Authors further applied the methods to height and obesity (BMI), and obtained the similar results.

This reviewer admits the nice statistical work done by the authors, but have empirical concerns on the interpretation of the results. The previous study on trans-ethnic PRS accuracy comparisons suggested that the accuracy differs in a trait-dependent manner. Martin AR et al (Nat Genet 2019) indicated that PRS accuracy between Europeans and East Asians were low in specific traits, such as obesity and type 2 diabetes. The conclusion of the current manuscript that LD and MAF differences can explain PRS accuracy reduction, may contradict the such previous results.

The simulation models adopted in this manuscript depends on the correlation (r_b) of causal SNP effect sizes between populations at equation 1. This reviewer is not sure this is true. When there exist ethnically different impacts of underlying biology on the trait, which is often suggested for obesity and type 2 diabetes, this hypothesis may not hold true.

Authors' suggestion could be one of the reasons to explain PRS accuracy reduction between the populations, but this reviewer is afraid that this manuscript could be overstating on one suggestion.

Authors may need to apply simulation results on a wide range of phenotypes.

Reviewer #2 (Remarks to the Author):

Wang et al have investigated the generalizability of polygenic scores across globally diverse populations, first deriving a theoretical model for assessing relative accuracy in two populations (importantly not using BLUP, which is most commonly modeled in theoretical work to date but is not typically used in humans), then empirically assessing generalizability using UK Biobank data for height and BMI. This problem is very important and timely as researchers and clinicians consider clinical applications of polygenic scores in humans. Specifically, a better understanding of genetic and environmental factors that bound relative prediction accuracy among diverse populations is critically important given Eurocentric biases in human genetics data. Furthermore, while many of the factors that impact polygenic score relative accuracy have been enumerated, this is the first study (to my knowledge) that has quantified the relative importance of each of these factors. Lastly, as sample sizes increase, this study provides a framework for evaluating how polygenic scores will generalize. This manuscript is also mostly well-written and contextualized. I am therefore highly supportive of publication and have listed comments to consider for improving the manuscript:

1. This study assumes relative homogeneity in GWAS in the theoretical setup, e.g. comparing Population 1 to Population 2. As the authors point out, large studies in many populations are hard to achieve, and thus trans-ethnic studies are increasingly popular. How do we use the theoretical model described here to predict relative accuracy in independent target populations when summary statistics are made available for multi-ethnic data (for example, reflecting a common current phenomenon: large numbers of Europeans, smaller fraction East Asians, small number of other minority populations)?
2. I found the loss of accuracy section of the results (pg 8-9) to be among the most interesting but took several reads to understand. I therefore thought Figure 4 could benefit from a schematic explaining the relationships between relative accuracy, loss of accuracy, and how LOA is partitioned into LD and MAF, genetic correlation and heritability, and negative selection. For example: "Our method predicts that LD and MAF explain 36.6% and 24.3% of the LOA of height- and BMI-PGS in participants of SAS; and up to 70.1% and 71.9% in participants of AFR for BMI and height, respectively." I wonder if the relative LOA explained by each genetic factor would be more meaningful. E.g., if the LOA for height is 44.7 in SAS and 84.3, it would be interesting to know whether the *relative* loss explained by LD and MAF is similar in both populations; this is currently hard to interpret because numbers are I think presented as an absolute loss.
3. The simulations and results are run using clumping. This is good, as it is most commonly applied in humans and easily to understand. However, methods such as LDpred can employ infinitesimal models and have been shown in some cases to be slightly more accurate within a population. How would these models impact simulation results of relative accuracy?
4. This paper is already very interesting but would be even more exciting to a broader audience if medically relevant traits such as heart disease and cancer traits (where PGS is closest to clinical applications) were also evaluated. To simplify evaluations, these could use GWAS summary statistics that exclude UK Biobank, enabling UKB to be used as a target/validation cohort. This may also be helpful to ensure the theoretical framework is clearly articulated for binary traits and because some traits often studied in these disease contexts (e.g. LDL) have genetic architectures that are very different from height and BMI.
5. Have the authors considered fine-mapping variants to see whether and by how much polygenic score relative accuracy improves, e.g. using FINEMAP, PAINTOR, or SuSiE? If so, how might this model be extended to include fine-mapping error (e.g. posterior inclusion probabilities) from fine-mapped credible sets to assess relative accuracies?
6. As GWAS increase in size, PGS become more accurate, and differences in relative accuracy should shrink. Some discussion around how this is laid out in the theoretical framework (I think through $\text{var}(\text{PGS})$) would be helpful.

Minor comments:

7. The equations use r for two different variables: genetic correlation and LD. Can the authors choose a different letter to avoid confusion from overloading this term? Relatedly, the authors define population as l for Equation 1 but never use it and could probably simplify.
8. Figures 1-2 describe "True," "Predicted," and "PGS-SNP," but not "Observed." This may be helpful to add.
9. De Vlaming et al, 2017 PLoS Genetics seems like relevant background.
10. The GCTA method is used to estimate heritability here. The authors may wish to comment on which methods for inferring heritability satisfy assumptions of the theoretical model.
11. This will not change central interpretations, so feel free to ignore, but the authors defined 4 populations using 3 PCs, the minimal viable number. There is still considerable population structure beyond this in the UK Biobank – was there a reason only 3 PCs were used for classification?

Reviewer #3 (Remarks to the Author):

Wang et al. present here a very well written paper, which studies the underlying cause for loss of prediction accuracies when applying polygenic trained on one population to another. The authors highlight several possible culprits, and work out an equation to estimate the expected loss of prediction accuracy under different parameters settings. The authors validate their model using simulations, and then study real traits (BMI and height) and fit their model to the observed loss of prediction accuracy. Although the paper is essentially focused on a single equation, I found the paper very interesting and insightful and I do not have any major substantial comments. I therefore recommend this paper to be accepted with some of the suggested revisions below that hopefully improve the manuscript and help broaden its appeal to the greater genetics community.

Comments:

- The model assumed is the standard additive genetic model, where both epistasis and dominance are ignored. However, one can imagine scenarios where the additive model is a reasonable assumption for analysis within a population, but may break down in cross population analyses. E.g., consider a model where neighboring epistatic interactions exist (say within a gene or exon). Under this model, the interacting variants will generally be in high LD, meaning that their effect will likely be captured well in an additive model. However, as soon as this LD breaks down, e.g. in a different population, their apparent marginal effects will change. Hence the effects between population will seem heterogenous.

In summary, there may be alternative explanations for what you see for BMI and height? I do not believe the epistasis model is more reasonable than the effects merely being heterogeneous, but I think it is worth mentioning as one possible explanation for what you observe for BMI and height. Also, Zuk et al. (PNAS 2012) suggested that epistatic interactions could be one possible reason for differences between additive twin and SNP heritabilities.

Finally, if there is regional epistasis, one would expect the extent of apparent effect heterogeneity to correlate with regional stability of LD between populations.

- Also, if you partition the genome into regions and order these by cross-population LD stability, is the PRS contribution from the stable parts more accurate than the other ones. Does this imply a possible weighting scheme to optimize the polygenic score? Perhaps that could be a future direction?

- Regarding capturing the signal from a "candidate causal variant". I don't understand why you restrict to $r^2 > 0.45$. I would think you could lower this and then capture more. I would be interested in seeing how lowering that threshold influences the results. In fact, is there a reason for why you cannot set the threshold to 0?

- Finally, I recommend that you apply your methods to more traits, ranging from polygenic to simpler monogenic disorders. It is unclear to me whether the results for BMI and height hold for other traits. Also, what happens if $S = -0.5$?

Minor comments:

- In the supplementary, I recommend deriving the equation in a slightly different order, for clarity. I suggest start by (11) before delving into equations 5-9, and instead deriving them as you need them. I find that a more natural order of deriving this.

- I would consider handling causal variants by allowing effect of 0, instead of always summing over $M^{(c)}$ causal variants. I think that could make the math simpler to read.

- p. 6 lines 184-188. The sentence is unclear.

Reviewer #1 (Remarks to the Author):

Authors conducted a simulation study empirically quantifying transferring accuracy of polygenic risk score (PRS) among ancestry divergent populations. They introduced a model which assesses effects of minor allele frequency (MAF) spectra and linkage disequilibrium (LD) differences. The simulation study demonstrated that the majority of the decreased accuracy of PRS between the populations were mostly explained by MAF and LD differences. The PRS of Europeans (EUR) were mostly transferrable to South Asians (SAS) or East Asians (EAS), when compared to Africans (AFR), reflecting distances between the populations. Authors further applied the methods to height and obesity (BMI), and obtained the similar results.

This reviewer admits the nice statistical work done by the authors, but have empirical concerns on the interpretation of the results. The previous study on trans-ethnic PRS accuracy comparisons suggested that the accuracy differs in a trait-dependent manner. Martin AR et al (Nat Genet 2019) indicated that PRS accuracy between Europeans and East Asians were low in specific traits, such as obesity and type 2 diabetes. The conclusion of the current manuscript that LD and MAF differences can explain PRS accuracy reduction, may contradict the such previous results.

We thank Reviewer #1 for their comment and apology if our conclusions seem to contradict results from Martin et al. (2019). In fact, they do not notably because we replicate many findings from the latter study (e.g. **Supplementary Figure 10**). We agree with the observation that the trans-ancestry prediction accuracy differs in a trait-dependent manner. We do now explicitly state that in our revised manuscript (Lines 117-118). In our model, trait-dependency is captured at different levels by the ratio of heritability (h^2_2/h^2_1) and the correlation of causal effects (previously denoted r_b but now ρ_b based on Reviewer #2's suggestions) between populations, which can be different across traits. Moreover, the local LD and frequency distribution of genome-wide significant SNPs may also differ between traits, which would lead our prediction of the relative accuracy attributable to MAF and LD to vary (in principle) between traits. To address the referee's concern and in accordance with other referees' comments we have (i) added more traits (lipids) and common diseases (asthma, hypertension and type 2 diabetes) in our real data analyses and (ii) also present results from different prediction methods (LDpred, SBayesR). These additions allow us to highlight differences between traits and more generally to illustrate the importance of the genetic architecture of traits and diseases on the trans-ancestry relative accuracy of PGS.

The simulation models adopted in this manuscript depends on the correlation (r_b) of causal SNP effect sizes between populations at equation 1. This reviewer is not sure this is true. When there exist ethnically different impacts of underlying biology on the trait, which is often suggested for obesity and type 2 diabetes, this hypothesis may not hold true.

We thank Reviewer #1 for their comment. We are not sure whether the referee questions the assumption that $r_b=1$ or more intrinsically the fact that the loss of accuracy depends on that correlation. Regarding the former interpretation, we do agree

that assuming a perfect correlation of causal effects between ancestries is a strong and very likely wrong assumption. In fact the presence of gene by gene or gene by environment interactions could lead to $|r_b| < 1$. Our model (Eq. 1) shows that the expected relative accuracy can be factorised as the product of two terms: (i) the contribution of genetic correlation and heritability differences and (ii) the contribution of LD and MAF. Because these two terms act multiplicatively, we could study them separately and thus focused our first simulations on the contribution of MAF and LD. This factorisation relies on the assumption that each SNP contributes equally to the genetic variance in each population. We did relax this assumption in our second simulation study (Impact of negative selection) and show that we can still well predict the contribution of MAF and LD despite a slight over-estimation. Following the referee's comment we made the following edits: (i) we added the following sentences: "As our main focus is to predict the fraction of the RA that can be attributed to alleles frequencies and LD differences between populations, we therefore assumed that effect sizes of causal variants are perfectly correlated across populations, i.e. $r_b = 1$ and that heritability is constant across populations, i.e. $h_2^2 = h_1^2 = h^2$. Note that in practice these assumptions are likely to be violated. However, our theoretical framework (Equations (1) and (2)) allows us to study the contribution of MAF and LD and that of heritability and genetic correlation separately. We relax this assumption below. (Lines 172-178)" and (ii) we underline in the Discussion the possibility that the correlation of causal effects between ancestries may not be independent of LD or MAF and therefore that a more complex model may be considered (Lines 395-398).

Authors' suggestion could be one of the reasons to explain PRS accuracy reduction between the populations, but this reviewer is afraid that this manuscript could be overstating on one suggestion. Authors may need to apply simulation results on a wide range of phenotypes.

We thank the referee for this comment. We have added more traits (and diseases) in our real data analysis and now highlight differences between traits regarding the expected relative contribution of MAF and LD to the prediction accuracy (Fig. 3).

Once more, we thank this referee for their comments and suggestions and hope that they will find our revised manuscript suitable for publication.

Reviewer #2 (Remarks to the Author):

Wang et al have investigated the generalizability of polygenic scores across globally diverse populations, first deriving a theoretical model for assessing relative accuracy in two populations (importantly not using BLUP, which is most commonly modelled in theoretical work to date but is not typically used in humans), then empirically assessing generalizability using UK Biobank data for height and BMI. This problem is very important and timely as researchers and clinicians consider clinical applications of polygenic scores in humans. Specifically, a better understanding of genetic and environmental factors that bound relative prediction accuracy among diverse populations is critically important given Eurocentric biases in human genetics data. Furthermore, while many of the factors that impact polygenic score relative accuracy have been enumerated, this is the first study (to my knowledge) that has quantified the relative importance of each of these factors. Lastly, as sample sizes increase, this study provides a framework for evaluating how polygenic scores will generalize. This manuscript is also mostly well-written and contextualized. I am therefore highly supportive of publication and have listed comments to consider for improving the manuscript:

We thank Reviewer #2 for their accurate summary of our study and for supporting the publication of our manuscript provided a few concerns are properly addressed. We provide below a point-by-point reply to each comment.

1. This study assumes relative homogeneity in GWAS in the theoretical setup, e.g. comparing Population 1 to Population 2. As the authors point out, large studies in many populations are hard to achieve, and thus trans-ethnic studies are increasingly popular. How do we use the theoretical model described here to predict relative accuracy in independent target populations when summary statistics are made available for multi-ethnic data (for example, reflecting a common current phenomenon: large numbers of Europeans, smaller fraction East Asians, small number of other minority populations)?

We thank the referee for this important question. Our approach could be extended straightforwardly to predict the relative accuracy from PGS based on SNPs effects estimated in a fixed-effect meta-analysis if the proportion of each ancestry is known and if the each study has homogenous ancestry. However, fixed-effect meta-analyses are rarely used for trans-ancestry studies and presence of proportion of ancestry can vary between SNPs if one the study contains admixed individuals. We now highlight this limitation and emphasise the need for more research in this area. (“In addition, our theoretical approach cannot be directly applied to predict the RA of PGS using SNP effects estimated from a trans-ancestry meta-analysis, although a straightforward extension could be derived for fixed-effects trans-ancestry meta-analyses. In practice, random effect models, which already accounts for allele frequency differences between ancestries, are often preferred over fixed-effects models. Therefore, we acknowledge that further theoretical work is required to address that specific question.”; Lines 375-380).

2. I found the loss of accuracy section of the results (pg 8-9) to be among the most interesting but took several reads to understand. I therefore thought Figure 4 could

benefit from a schematic explaining the relationships between relative accuracy, loss of accuracy, and how LOA is partitioned into LD and MAF, genetic correlation and heritability, and negative selection. For example: “Our method predicts that LD and MAF explain 36.6% and 24.3% of the LOA of height- and BMI-PGS in participants of SAS; and up to 70.1% and 71.9% in participants of AFR for BMI and height, respectively.” I wonder if the relative LOA explained by each genetic factor would be more meaningful. E.g., if the LOA for height is 44.7 in SAS and 84.3, it would be interesting to know whether the *relative* loss explained by LD and MAF is similar in both populations; this is currently hard to interpret because numbers are I think presented as an absolute loss.

We thank the referee for their comment. We apologise for not making this section clearer before. As suggested by the referee, we have now added a schematic figure (**Supplementary Figure 12**) to help the reader better interpret our results. We now also discuss the ratio of predicted loss of accuracy (1-predicted RA) from LD and MAF differences over the observed LOA (1-observed RA). For height that ratio is ~23.6% in SAS vs ~71.5% in AFR. This observation suggests that LD and MAF differences contribute more to the LOA in AFR than they do in SAS. We present similar results and discussions for other traits in our revised manuscript (Lines 307-327).

3. The simulations and results are run using clumping. This is good, as it is most commonly applied in humans and easily to understand. However, methods such as LDpred can employ infinitesimal models and have been shown in some cases to be slightly more accurate within a population. How would these models impact simulation results of relative accuracy?

We thank the referee for this question and suggestion, upon which we have re-analysed our simulated data using LDpred and SBayesR. We now report these results in **Supplementary Figure 13**. Overall, we confirm that genome-wide significant SNPs alone do not yield an optimal accuracy and show that a method like SBayesR, which can adapt to different genetic architectures not only performs better within participants of European ancestry (as previously reported; Lloyd-Jones et al. *Nat. Comm.* 2019) but also suffers from a smaller loss of accuracy in non-European ancestries. Interestingly, we found that the improvement of LDpred or SBayesR relative to our clumping strategy is near proportional in EUR and non-EUR ancestries (**Supplementary Figure 13-C,D**). This translates as a similar relative accuracy across methods, although we found SBayesR to yield the largest RA across scenarios. We have added these new elements to our discussion (Lines 371-382).

4. This paper is already very interesting but would be even more exciting to a broader audience if medically relevant traits such as heart disease and cancer traits (where PGS is closest to clinical applications) were also evaluated. To simplify evaluations, these could use GWAS summary statistics that exclude UK Biobank, enabling UKB to be used as a target/validation cohort. This may also be helpful to ensure the theoretical framework is clearly articulated for binary traits and because some traits often studied in these disease contexts (e.g. LDL) have genetic architectures that are very different from height and BMI.

We thank the referee for this important comment. We have now added analyses of additional traits (lipids) and three diseases (type 2 diabetes, asthma and hypertension)

common in the UK Biobank to widen the scope of our real data analysis (Lines 294-307; **Fig. 3**). We note that disease diagnostic criteria and severity heterogeneity between cohorts in meta-analyses may potentially contribute to the loss of accuracy. In this study, we wished to minimise the impact of these factors, and therefore all discovery GWAS were performed within the UK Biobank as done in the previous version of our manuscript.

5. Have the authors considered fine-mapping variants to see whether and by how much polygenic score relative accuracy improves, e.g. using FINEMAP, PAINTOR, or SuSiE? If so, how might this model be extended to include fine-mapping error (e.g. posterior inclusion probabilities) from fine-mapped credible sets to assess relative accuracies?

We thank Reviewer #2 for this question and suggestion. To answer the referee's first question, we re-analysed our simulated data using the software FINEMAP. We considered two scenarios: scenario 1 with $M_C=1,000$ causal variants and a heritability $h^2=0.5$, and scenario 2 with $M_C=10,000$ and $h^2=0.25$. We defined the genomic region for fine-mapping as the region within 500Kb of the lead genome-wide significant SNPs selected from LD clumping. We used the shotgun stochastic search algorithm (option `--sss`) to select credible sets, within which posterior conditional effect sizes were estimated. Next, we used posterior conditional SNP effects from FINEMAP to calculate PGSs and evaluated accuracy and relative accuracy of the latter predictor. As sample size of LD reference has been previously shown to play a role in the predictive power of PGS methods (Lloyd-Jones et al. 2019), we ran FINEMAP using both ~ 500 participants of European ancestry from the 1000 genomes project and 50,000 unrelated participants of British ancestry from the UKB. Overall, the conclusion from this analysis is that fine-mapping based PGS perform better across ancestry if the LD reference is large enough (**Fig. R1** below). However, this analysis is partial as it only considered one fine-mapping algorithm, one set of whole genome-sequence data, etc. We believe that this question should be further investigated in a follow up study and therefore do not report these partial results in our manuscript.

Regarding, the referee's second question, fine-mapping results can straightforwardly be integrated with our method as an alternative to our definition of candidate causal variants. Importantly, fine-mapping posterior probabilities can be used as weights for candidate causal SNPs, which our current heuristics cannot do. However, it is also worth noting an important advantage of our heuristic method, which is that it utilises WGS data and therefore candidate causal variants that are not present in the GWAS summary statistics may still be used for inference. In contrast standard fine-mapping tools are limited by the resolution of GWAS summary statistics, although that resolution can be improved using summary statistics imputation at the expense of an extra-step.

Fig. R1. Trans-ancestry absolute and relative prediction accuracy of fine-mapping based polygenic score methods.

6. As GWAS increase in size, PGS become more accurate, and differences in relative accuracy should shrink. Some discussion around how this is laid out in the theoretical framework (I think through $\text{var}(\text{PGS})$) would be helpful.

We thank the referee for this comment. We are not quite sure what the referee means by differences in relative accuracy given that the relative accuracy is already measuring a difference. As suggested by the referee, we ran additional simulations, in which we varied the discovery sample size between 100,000 and 300,000. As expected, we found that accuracy of PGS increased in all ancestries with sample size of the discovery GWAS. However, that improvement also was proportional across ancestries such that the relative accuracy remains constant (**Supplementary Figure 8**). This observation is consistent with other analyses we have performed showing that the improvement in prediction accuracy is nearly proportional across all ancestries when using more powerful polygenic scoring methods like LDpred or SBayesR (**Supplementary Figure 13**).

Minor comments:

7. The equations use r for two different variables: genetic correlation and LD. Can the authors choose a different letter to avoid confusion from overloading this term?

We thank the referee for this suggestion. We now use “rho” to denote the correlation of causal effects.

Relatedly, the authors define population as l for Equation 1 but never use it and could probably simplify.

We thank the referee for this observation. We have removed mention to population l and explicitly use the notations $p_{k,1}$ and $p_{k,2}$ for minor allele frequencies at SNP k in populations 1 and 2.

8. Figures 1-2 describe “True,” “Predicted,” and “PGS-SNP,” but not “Observed.” This may be helpful to add.

We thank the referee for this point. We have now added the description for the “Observed” label in our figures. We changed the labels “True”, “Predicted,” and “PGS-SNP,” to “Predicted1”, “Predicted2” and “Predicted3” and explained them in the figure legend for clarity.

9. De Vlaming et al, 2017 PLoS Genetics seems like relevant background.

We thank the referee for this point. We have now added a reference to de Vlaming et al. (2017) as a special case of our Equation (1).

10. The GCTA method is used to estimate heritability here. The authors may wish to comment on which methods for inferring heritability satisfy assumptions of the theoretical model.

Equation (1) is based on the assumption that each SNP contributes equally to the trait heritability regardless of local LD and frequency. This is the same as the standard assumption underlying the GREML model implemented in GCTA (Yang et al. *AJHG*, 2011). Note that we challenged those assumptions in our second simulation studies where we introduced a relationship between effect sizes and allele frequencies. However, as mentioned in our response to the Editor, we have now removed these estimations of heritabilities from the revised manuscript.

11. This will not change central interpretations, so feel free to ignore, but the authors defined 4 populations using 3 PCs, the minimal viable number. There is still considerable population structure beyond this in the UK Biobank – was there a reason only 3 PCs were used for classification?

We thank the referee for this question. We acknowledge that our choice of 3 PCs is minimal. To answer this referee’s comment, we varied the number of PCs used to call ancestry from 3 to 10 and thus obtained 8 classifications of UKB participants into four classes: EUR, SAS, EAS and AFR. We then performed a pair-wise comparison of these classifications by calculating the classification error rates (R package “mclust”, function: “classError”), which measures the percentage of classification errors if one partition is used a reference. Over these $8 \times 7 / 2 = 28$ comparisons, we found a classification error $< 0.4\%$, suggesting that a fairly small fraction of participants is reclassified as we increase the number of PCs. These additional analyses suggest a little impact on our results and interpretation.

Reviewer #3 (Remarks to the Author):

Wang et al. present here a very well written paper, which studies the underlying cause for loss of prediction accuracies when applying polygenic trained on one population to another. The authors highlight several possible culprits, and work out an equation to estimate the expected loss of prediction accuracy under different parameters settings. The authors validate their model using simulations, and then study real traits (BMI and height) and fit their model to the observed loss of prediction accuracy. Although the paper is essentially focused on a single equation, I found the paper very interesting and insightful and I do not have any major substantial comments. I therefore recommend this paper to be accepted with some of the suggested revisions below that hopefully improve the manuscript and help broaden its appeal to the greater genetics community.

We thank this referee for acknowledging the quality of our manuscript and supporting its publication provided some revisions are properly addressed. We provide below a point-by-point reply to each point made by this referee.

Comments:

1. The model assumed is the standard additive genetic model, where both epistasis and dominance are ignored. However, one can imagine scenarios where the additive model is a reasonable assumption for analysis within a population, but may break down in cross population analyses. E.g., consider a model where neighboring epistatic interactions exist (say within a gene or exon). Under this model, the interacting variants will generally be in high LD, meaning that their effect will likely be captured well in an additive model. However, as soon as this LD breaks down, e.g. in a different population, their apparent marginal effects will change. Hence the effects between population will seem heterogenous.

In summary, there may be alternative explanations for what you see for BMI and height? I do not believe the epistasis model is more reasonable than the effects merely being heterogeneous, but I think it is worth mentioning as one possible explanation for what you observe for BMI and height. Also, Zuk et al. (PNAS 2012) suggested that epistatic interactions could be one possible reason for differences between additive twin and SNP heritabilities.

Finally, if there is regional epistasis, one would expect the extent of apparent effect heterogeneity to correlate with regional stability of LD between populations.

We thank the referee for their thorough analysis of the possible contribution of within-locus (or cis-) epistasis to our observation. We now mention in our discussion (Line 379-382) that such types of epistasis may contribute to explaining our observations. However, we note here that there has been little quantification to date of the magnitude of its contribution to genetic variance.

2. Also, if you partition the genome into regions and order these by cross-population LD stability, is the PRS contribution from the stable parts more accurate than the other

ones. Does this imply a possible weighting scheme to optimize the polygenic score? Perhaps that could be a future direction?

We thank the referee for this suggestion. We agree that investigating this question is critical and believe that it should deserve a separate publication, which goal would be to develop a new polygenic scoring method that maximises trans-ancestry prediction accuracy. As suggested by this referee, we looked at the observed prediction accuracy of PGS based on SNPs which local LD in non-European ancestries differ little from that of European ancestry. For each SNP included in the polygenic score (or PGS-SNPs), we measured the consistency of local LD between European ancestry and other non-European ancestries using the following statistic: $\ell_{LD,k} = \overline{r_{k,1}r_{k,2}}/\overline{r_{k,1}^2}$ [details from main text: “We use the notation $\overline{r_{k,1}^2}$ to denote the mean squared correlation of allele counts between the k^{th} PGS-SNP and all candidate causal SNPs. Similarly, we define $\overline{r_{k,1}r_{k,2}}$ as the mean of $r_{jk,1}r_{jk,2}$ ’s between the k^{th} PGS-SNP and all candidate causal SNPs”]. Note that large values of $\ell_{LD,k}$ indicate high correlation of local LD pattern between European and non-European ancestries.

We next partitioned PGS-SNPs into two groups: Group1 containing SNPs with $\ell_{LD,k}$ larger than the median of $\ell_{LD,k}$ ’s and Group2 containing SNPs with $\ell_{LD,k}$ lower than the median of $\ell_{LD,k}$ ’s. We also used the $\ell_{LD,k}$ ’s as additional weights (i.e. SNPs are weighted by $\ell_{LD,k} \times \hat{\beta}_k$) to construct the weighted PGS (denoted in **Fig. R2** below as “Weighted”).

Fig. R2. Prediction accuracy of PGS from SNPs stratified (or weighted) according to the consistency of local LD between European and non-European ancestries.

As shown in **Fig. R2** above and expected by Reviewer #3, we find that PGS based on SNPs from Group 1 perform significantly better than those from Group 2. However, we

do not see a significant improvement of our weighting strategy. These observations underscore that there is a potential to improve accuracy of PGS across ancestries from properly modelling local LD differences but also highlight that better approaches must be developed in future studies.

3. Regarding capturing the signal from a “candidate causal variant”. I don’t understand why you restrict to $r^2 > 0.45$. I would think you could lower this and then capture more. I would be interested in seeing how lowering that threshold influences the results. In fact, is there a reason for why you cannot set the threshold to 0?

We thank the referee for this comment. We apologise if that part of our manuscript was not clearer. To model the loss of accuracy, our reasoning is based on using causal variants as anchors. However, as in practice causal variants are largely unknown, we used two arguments to narrow them down. First, we focused on genome-wide significant (GWS) SNPs and secondly, we used results from the extensive simulation study of Wu et al. (2017) to elicit candidate causal variants as variants located with < 100 kb and such that their LD r^2 with the GWS SNP is > 0.45 . Because these variants are GWS they should be well tagging causal variants hence r^2 between GWS SNPs and causal variants must be large enough. Assuming a too small r^2 would imply that causal effects are disproportionately large. For example, if a GWS explain $q^2(\text{SNP}) = 1\%$ of trait variance and if it only tags on causal SNP then $q^2(\text{SNP}) = r^2(\text{SNP,causal}) * q^2(\text{causal}) / [1 - q^2(\text{causal})] = 1\%$. In other words, we expect in this case the variance explained by the underlying causal variant to be at most $(0.01/0.45) / [1 + (0.01/0.45)] \sim 2.2\%$. Putting that threshold to say 0.01, will be unreasonable because that would imply that this causal variant could potentially explain 100% of phenotypic variance. Finding the right threshold is a difficult question, which fortunately Wu and colleagues have addressed previously.

To further illustrate our point, we have varied the threshold for r^2 from 0 to 0.6 in our simulations (see below **Fig. R3**). We find that setting $r^2 > 0$ leads to predict a too large loss of accuracy, while setting that threshold above 0.6 leads to an underestimation of that loss. Note that the method denoted in our manuscript as “naïve” corresponds to a threshold above 0.99 and is shown to strongly under-estimate the loss of accuracy (**Fig. 1** in the manuscript).

Our strategy to determine “candidate” causal variants has limitations and we already discussed and acknowledged them in the manuscript. Alternatively, standard fine-mapping methods provide a more rigorous model-based approach which is easily justifiable. Yet, these methods also bear some heuristics elements as users need to provide cut-off thresholds (e.g. for posterior probabilities) or sizes of genomic windows (see reply to Reviewer #2). One important advantage of our heuristic method is that it utilises WGS data and therefore candidate causal variants that are not present in the GWAS summary statistics may still be used for inference. In contrast standard fine-mapping tools are limited by the resolution of GWAS summary statistics.

Fig. R3. Absolute and relative accuracy of PGS using different linkage disequilibrium threshold to define candidate causal variants.

4. Finally, I recommend that you apply your methods to more traits, ranging from polygenic to simpler monogenic disorders. It is unclear to me whether the results for BMI and height hold for other traits. Also, what happens if $S=-0.5$?

We thank the referee for their suggestion, which is consistent with those from the two other referees. Accordingly, we have added analyses of blood lipids (less polygenic continuous traits) and three common diseases with varied levels of polygenicity. Finally, we did address the referee's last question in our second simulation (**Fig. 2** in the main text). We find that our approach which implicitly assumed $S=-1$ would lead to overestimate the loss of accuracy but the magnitude of that of that bias decrease with polygenicity. We are not sure what the referee means by "what happens if $S=-0.5$ " given that we dedicated a section quantifying the impact of misspecification of the S parameter. If the referee is asking why we chose values around -0.5 , then the answer to that question is that estimates of S across a large number of traits (Zeng et al. 2019; biorxiv) were found to vary between -0.25 and -0.75 . So we took the middle of that interval as our reference point. [Zeng et al. (2019): Bayesian analysis of GWAS summary

data reveals differential signatures of natural selection across human complex traits and functional genomic categories].

Minor comments:

A) In the supplementary, I recommend deriving the equation in a slightly different order, for clarity. I suggest start by (11) before delving into equations 5-9, and instead deriving them as you need them. I find that a more natural order of deriving this.

We thank the referee for this comment. This is now addressed.

B) I would consider handling causal variants by allowing effect of 0, instead of always summing over M^c causal variants. I think that could make the math simpler to read.

We thank the referee for this comment. In fact, we do allow certain causal variants to have an effect of 0. However, our theory predicts the expected relative accuracy and therefore need to take the expectation over the distribution of causal (squared) effects, which is never 0.

C) p. 6 lines 184-188. The sentence is unclear.

We thank the referee for this comment. We have now rephrased the sentence.

REVIEWERS' COMMENTS:

Reviewer #1 (Remarks to the Author):

Wang et al. revised their manuscript on trans-ethnic applicability of PRS. Authors carefully responded the reviewer's comments, including application to wider ranges of phenotypes.

Reviewer #2 (Remarks to the Author):

The authors were very responsive to my comments and conducted additional work with thoughtful responses to them. I have no further comments, aside from the fact that this careful work on an important topic will make an excellent publication.

Reviewer #3 (Remarks to the Author):

The authors have addressed all of my comments well, and I have no further comments. I believe this work is a valuable contribution to the field.